# GROKFAST:
# GRADIENT FILTERS FOR FASTER GROKKING

## ABSTRACT

One puzzling artifact in machine learning, dubbed *grokking*, refers to the case where a model exhibits delayed generalization after numerous training iterations after nearly perfect overfitting. Focusing on the long delay itself on behalf of machine learning practitioners, our primary goal is to accelerate the generalization of a model under the grokking phenomenon. By regarding a series of gradients of a parameter over training iterations as a random signal over time, we can spectrally decompose the parameter trajectories under gradient descent into two components: the fast-varying, overfitting-yielding component, and the slow-varying, generalization-inducing component. This analysis allows us to accelerate the grokking phenomenon more than $\times 50$ with only a few lines of code that amplifies the slow-varying components of the gradients. The experiments show that our algorithm applies to diverse tasks involving images, languages, and graphs, enabling the practical availability of this peculiar artifact of sudden generalization. Moreover, we reinterpret momentum hyperparameters in gradient-based optimizers as low-pass filters with size-1 windows. This bridges between optimization and classical signal processing literature, suggesting a new type of optimzers augmented with frequecy-domain filters.

## 1 INTRODUCTION

Grokking is a recently discovered phenomenon in which generalization is achieved long after a model overfits the training data. The phenomenon was first reported by Power et al. (2022) for a two-layer Transformer (Vaswani et al., 2017) trained using a simple algorithmic dataset. Later, Liu et al. (2022b) has shown that similar artifacts are observed for various model architectures trained with a variety of datasets, including images, languages, and graphs. Many theory-oriented works have tried to justify the effect by relating the grokking phenomenon to the previously known double descent phenomenon (Davies et al., 2023; Huang et al., 2024), yet its cause and sufficient conditions have not been fully characterized.

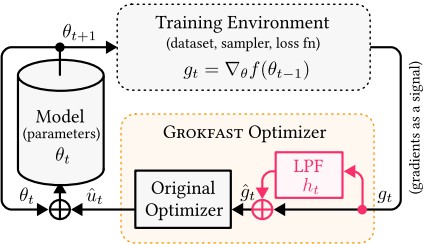

Figure 1: GROKFAST interprets training dynamics of a network parameter as a stochastic signal and amplify the low frequency variations for faster grokking.

Apart from theoretical studies, this work takes a practitioner's standpoint to take advantage of the grokking phenomenon. In previous reports on grokking (Power et al., 2022; Liu et al., 2022b), generalization is observed only after more than hundred times the training iterations after overfitting. This high demand for computational resources means less practical appeal to general machine learning practitioners, who are often under dire resource constraints. Therefore, achieving faster generalization in those overfitting systems is a necessary step to fully exploit the potential of this unusual behavior. From this perspective, the goal of this work is to accelerate the grokking phenomenon.

In the example training curve of a model under grokking in Figure 2, the dynamics of the validation loss are a few orders of magnitude slower than the dynamics of the training loss. The change in losses is a direct consequence of the change in the parameter values throughout the training session. Hence, Figure 2 suggests that parameter updates under grokking take effect in two different timescales: the

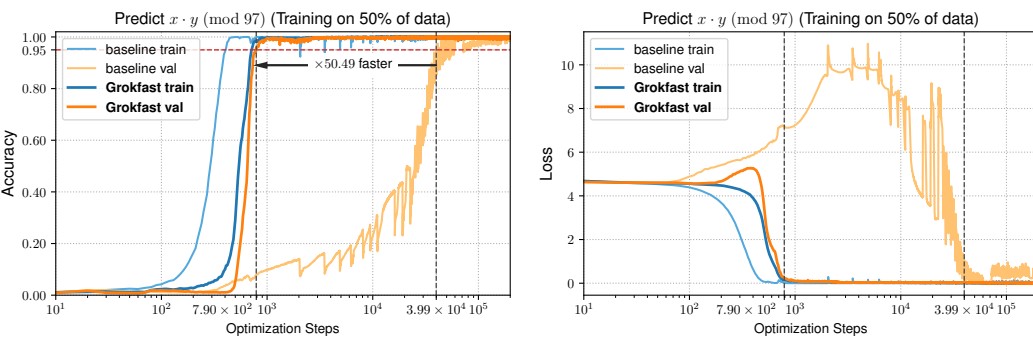

(a) Accelerated grokking with GROKFAST-MA.   (b) Corresponding loss curves.

Figure 2: **Accelerating generalization of a model under grokking phenomenon.** Our GROKFAST is a class of simple algorithmic modifications upon existing optimizers to pull forward the time of event of sudden generalization after overfitting, also known as the *grokking* phenomenon.

fast-varying component of parameter updates contributes to the rapid overfitting, and the slow-varying component contributes to the slow generalization.

We begin by treating the change in value $u(t)$ of each model parameter $\theta$ over the training iteration $t$ of an optimizer as a discrete (random) signal over time. As the optimizer iterates the training data, the value of each parameter $\theta(t)$, the loss $l(t)$, and its gradient $g(t) := \partial l(t)/\partial \theta(t)$ drift with respect to the guidance from a sequence of randomly selected mini-batches sampled at each iteration $t$:

$$\theta(t + 1) = \theta(t) + u(g(t), t) = \theta(0) + \sum_{\tau=0}^{t} u(g(\tau), \tau).$$ (1)

The parameter update function $u(t) = u(g(t), t) = \theta(t + 1) - \theta(t)$ provides a simple abstraction of the underlying optimizer. This notation can describe different instances of iterative optimizers, including SGD with various hyperparameters, e.g., learning rate, and momentum.

Treating the optimization process as a collection of discrete random signals $u(t)$ allows us to consider its dual representation $U(\omega)$ in the *frequency domain*. Taking the discrete-time Fourier transform $\mathcal{F}$ of $u(t)$ with respect to the training iteration $t$, we obtain the spectral representation of the sequence of changes of a specific parameter $\theta$:

$$U(\omega) = \mathcal{F}\{u(t)\} = \sum_{t=0}^{T} u(t)e^{-i\omega t},$$ (2)

where $T$ is the total number of training iterations in this specific training session. The slow-varying part of the parameter updates $u(t)$ which is related to the delayed generalization under grokking is then the low-frequency component of the dual $U(\omega)$. Furthermore, for gradient-based optimizers, the parameter update $u(g(t), t)$ is determined by the sample gradients $g(t)$ generated from back-propagation performed at each training step $t$. Therefore, we can associate slow generalization under grokking to the low-frequency part of gradient signals $G(\omega) = \sum_{t=0}^{T} g(t)e^{-i\omega t}$. This leads to our main hypothesis: *amplifying the low-frequency component of $G(\omega)$ accelerates the speed of generalization under grokking phenomenon*. This is to facilitate the research on grokking and to broaden our understanding on optimization dynamics of grokked models.

In the following sections, we empirically demonstrate this claim with a simple low-frequency gradient amplifier in various scenarios. These include tasks involving various network architectures including Transformers (Vaswani et al., 2017), MLPs, RNNs and (Graph-)ConvNets and diverse datasets such as algorithmic data, images, languages, and graphs that are treated to exhibit the grokking phenomenon (Liu et al., 2022b). Our method is simple, taking only a few lines of additional code, and is applicable to most machine learning frameworks such as PyTorch (Paszke et al., 2019), with ×50 faster exhibition of grokking as shown in Figure 2.

| **Algorithm 1** GROKFAST-EMA (GROKFAST). | **Algorithm 2** GROKFAST-MA |
|---|---|
| 1: **Param:** scalar momentum $\alpha$, factor $\lambda$. | 1: **Param:** window size $w$, scalar factor $\lambda$. |
| 2: **Input:** initial parameters $\theta_0$, stochastic objective function $f(\theta)$, optimizer's parameter update $u(g,t)$ from gradient $g$ at timestep $t$. | 2: **Input:** initial parameters $\theta_0$, stochastic objective function $f(\theta)$, optimizer's parameter update $u(g,t)$ from gradient $g$ at timestep $t$. |
| 3: **begin:** $t \leftarrow 0$; $\mu \leftarrow \theta_0$: EMA of gradients. | 3: **begin:** $t \leftarrow 0$; $Q \leftarrow \text{Queue(capacity} = w)$ |
| 4: **while** $\theta_t$ not converged **do** | 4: **while** $\theta_t$ not converged **do** |
| 5:    $t \leftarrow t + 1$ | 5:    $t \leftarrow t + 1$ |
| 6:    $g_t \leftarrow \nabla_\theta f(\theta_{t-1})$: Calculate gradients. | 6:    $g_t \leftarrow \nabla_\theta f(\theta_{t-1})$: Calculate gradients. |
| 7:    $\mu \leftarrow \alpha\mu + (1-\alpha)g_t$: Calculate EMA. | 7:    $\text{Insert}(Q, g_t)$: Insert gradients to $Q$. |
| 8:    $\hat{g}_t \leftarrow g_t + \lambda\mu$: Filter gradients. | 8:    $\hat{g}_t \leftarrow g_t + \lambda \cdot \text{Avg}(Q)$: Filter gradients. |
| 9:    $\hat{u}_t \leftarrow u(\hat{g}_t, t)$: Calculate update. | 9:    $\hat{u}_t \leftarrow u(\hat{g}_t, t)$: Calculate update. |
| 10:    $\theta_t \leftarrow \theta_{t-1} + \hat{u}_t$: Update parameters. | 10:    $\theta_t \leftarrow \theta_{t-1} + \hat{u}_t$: Update parameters. |
| 11: **end while** | 11: **end while** |

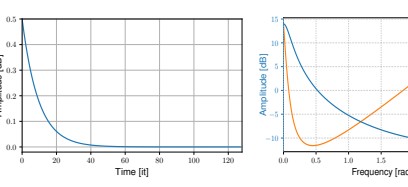 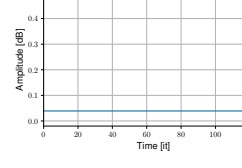 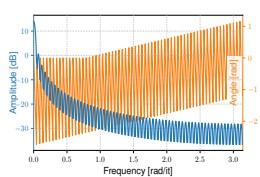

| (a) Time plot of EMA. | (b) Freq. plot of EMA. | (c) Time plot of MA. | (d) Freq. plot of MA. |

Figure 3: **Time and frequency domain plots of the gradient filters.** Figures (a, b) and (c, d) depict the impulse responses and the transfer functions of the filters of Algorithm 1 and 2, i.e., the EMA and the MA filters $h(t)$. We treat training iterations as discrete timesteps.

## 2 AMPLIFYING THE LOW-FREQUENCIES OF THE STOCHASTIC GRADIENTS

### 2.1 FILTERING TIME-VARYING GRADIENT SIGNALS IN FREQUENCY DOMAIN

Amplifying the low-frequencies of the gradients $g(t)$ can be achieved by adding a low-pass filtered signal $g(t)$ to itself. Let $h(t)$ be a discrete-time low-pass filter (LPF) defined over the training iteration $t$. For simplicity, we assume a univariate time-invariant low-pass filter $h(t)$ uniformly applied across every model parameter $\theta$. Using a convolution operator $*$, we denote the modified gradient $\hat{g}(t)$ as:

$$\hat{g}(t) = g(t) + h(t) * g(t), \tag{3}$$

which can then be plugged into the parameter update function $u$ of the optimizer:

$$\hat{u}(t) = u(\hat{g}(t), t) = u(g(t) + h(t) * g(t), t). \tag{4}$$

In the dual domain, equation equation 3 is equivalent to:

$$\hat{G}(\omega) = G(\omega) + H(\omega)G(\omega) = (1 + H(\omega))G(\omega), \tag{5}$$

where $H(\omega) = \sum_{t=0}^{T} h(t)e^{-i\omega t}$ is the transfer function of the filter $h(t)$. Our goal can therefore be restated as to design a filter $h(t)$ with low-pass characteristics in its transfer function $H(\omega)$.

### 2.2 GROKFAST WITH EMA GRADIENT FILTER

For the demonstration of our initial claim, we first take the simplest strategy: the LPF $h(t)$ is an exponential moving average (EMA) with momentum $\alpha$. The impulse response of the filter becomes:

$$h(t) = \lambda(1-\alpha)\sum_{\tau=0}^{t} \alpha^\tau \delta(t-\tau) = \lambda\alpha^t(1-\alpha), \tag{6}$$

where $\delta(t)$ is the discrete unit impulse at the origin. This filter also has two hyperparameters: the scalar factor $\lambda$ and the scalar momentum $\alpha$. The time and the frequency responses of the filters are shown in Figure 3. The resulting Algorithm 1 is implemented by inserting a single line of code as shown in Appendix C.

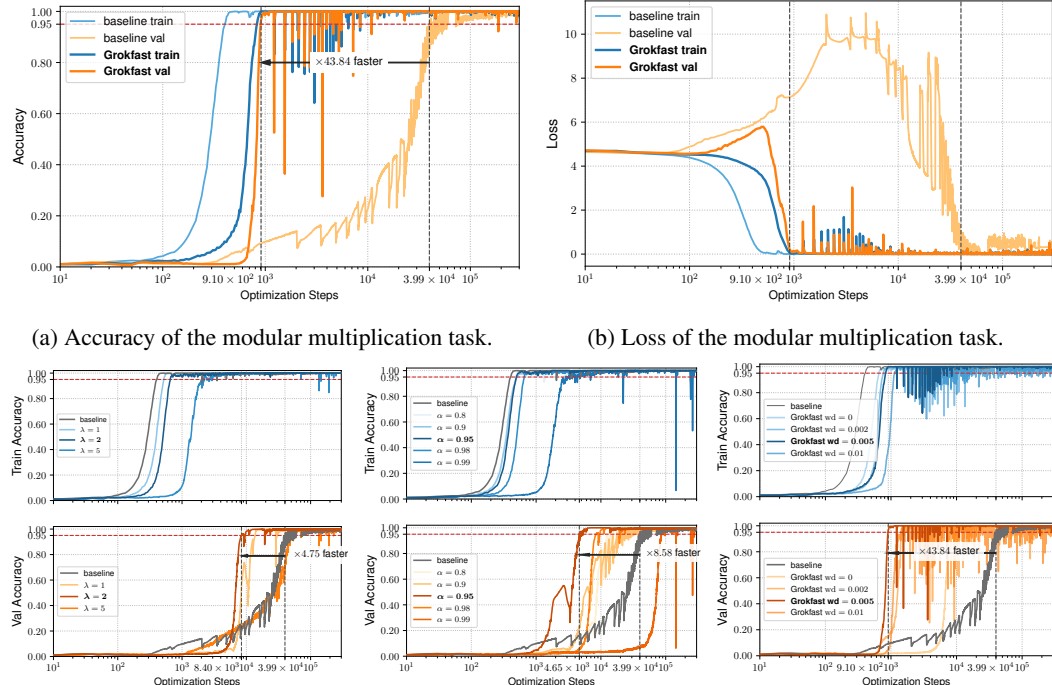

(a) Accuracy of the modular multiplication task.

(b) Loss of the modular multiplication task.

(c) Accuracy w.r.t. amplifier gain $\lambda$.  (d) Accuracy w.r.t. momentum $\alpha$.  (e) Accuracy w.r.t. weight decay.

Figure 4: **Acceleration of delayed generation with GROKFAST-EMA (GROKFAST).** The task is the same modular multiplication as in Figure 8. The amount of acceleration relies on three hyperparameters, the amplifier gain $\lambda$, the window size $w$, and the weight decay wd. Figures 4a and 4b use $\alpha = 0.98$, $\lambda = 2.0$, and wd $= 0.005$. Figures 4c and 4d show acceleration results when wd $= 0$. Figures 4c, 4d, and 4e use the same set of hyperparameters unless specified otherwise.

## 3 EXPERIMENT

Although the grokking phenomenon was first reported in the algorithmic dataset, Omnigrok (Liu et al., 2022b) shows that such behavior can also be observed in a diverse set of tasks with larger and more complex datasets. This section validates the efficacy of our accelerating algorithm, GROKFAST, for those various tasks and models that exhibit the grokking phenomenon.

### 3.1 ALGORITHMIC DATA

We first train the same task with the same model as in the first report on grokking (Power et al., 2022) using our new Algorithm 1. Specifically, we train a two-layer decoder-only Transformer (Vaswani et al., 2017) for a modular binary multiplication $x \cdot y \pmod{97}$, the same task where the grokking phenomenon is firstly observed (Power et al., 2022). Figure 4 reveals that the simplest implementation of GROKFAST with exponential moving average effectively captures the slow variation of the gradients necessary for accelerating the delayed generalization. Under the grokking phenomenon, the validation loss of the model first increases before it decreases again later during the late generalization stage as depicted in Figure 4b (baseline). This implies that GROKFAST effectively keeps the model parameters closer to the global optimum before the generalization happens. This difference in training dynamics are revisited in Section 5 with more visualization.

We also conduct ablation studies to find out the effect of hyperparameters $\lambda$, $\alpha$, and weight decay for our GROKFAST algorithm with an EMA filter. The optimal hyperparameters are found with grid search. Figures 4c through 4e summarizes the results. Recalling that our main idea is at the design of a low-pass filter, the momentum parameter $\alpha$ of Algorithm 1 (as well as the window size parameter $w$ of Algorithm 2) is equivalent to the cutoff frequency of the underlying filter. Experiments in Figures 4c through 4e as well as those in Figures 8 and 11 show that there exists a sweet spot in cutoff frequency that corresponds to the generalization-inducing gradient signal. From our empirical studies,

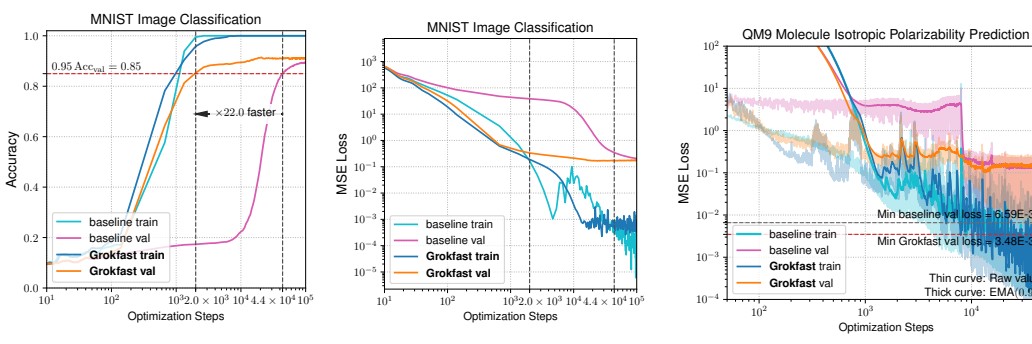

(a) Accuracy of MNIST.

(b) Loss of MNIST.

Figure 6: QM9 dataset results with a GCNN. GROKFAST achieves faster and better convergence.

Figure 5: MNIST results with a three-layer MLP. Grokking phenomenon is almost gone with proper hyperparameters.

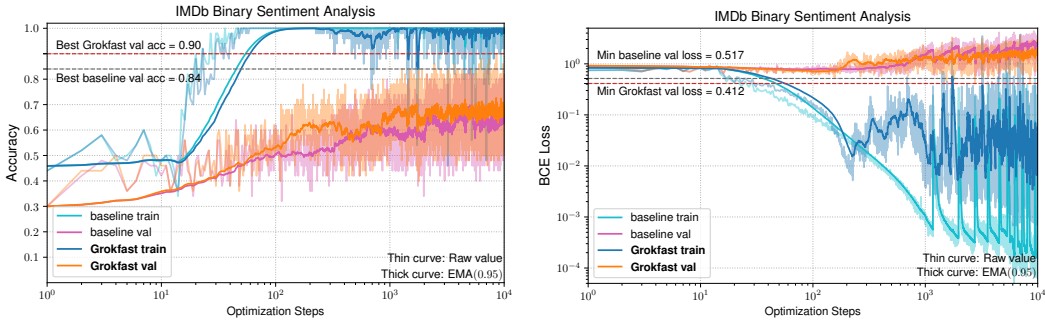

(a) Accuracy on IMDb sentiment analysis.

(b) Loss on IMDb sentiment analysis.

Figure 7: IMDb results with a two-layer LSTM. LSTM exhibits grokking phenomenon if the training begins with larger weight values at initialization. GROKFAST algorithm produces faster generalization with higher validation accuracy and lower validation loss. We show the exponential moving average (momentum 0.95) of the curves as thick lines for clearer visualization of the trend.

we recommend $\lambda \in [0.1, 5]$ and $\alpha \in [0.8, 0.99]$. The weight decay is, like in typical optimization problems, dependent on the task of interest.

## 3.2 MNIST

Besides the simple algorithmic reasoning task, where the data is relatively simple, Liu et al. (2022b) report the similar delayed generalization can also be observed in many conventional tasks if the model goes through a special treatment. To demonstrate the generalizability of our GROKFAST modification of the optimization process, we try to accelerate the speed of generalization under those reported models and tasks. The first is a three-layer ReLU-MLP trained for MNIST classification task (Deng, 2012) which exhibits the grokking phenomenon. Figure 5 summarizes the results, showing that our Algorithm 1 successfully accelerate the delayed generalization. With $\alpha = 0.8$, $\lambda = 0.1$, and $\mathrm{wd} = 2.0$, the delay until grokking is reduced by $\times 22.0$. Moreover, the final evaluation accuracy becomes higher from 89.8% to 91.2%.

## 3.3 QM9

In the next experiment, we train a graph convolutional neural network (GCNN) trained for a molecule dataset QM9 (Ruddigkeit et al., 2012; Ramakrishnan et al., 2014). Since this task does not have an accuracy measure to compare the speed of convergence, we instead compare the convergence speed of the validation loss. With the same setup as in Omnigrok (Liu et al., 2022b), elaborated in Appendix B, we apply Algorithm 1 with $\alpha = 0.9$, $\lambda = 1.0$, and $\mathrm{wd} = 0.01$ to obtain the results in Figure 6. The validation loss drops faster *and* by a larger margin under GROKFAST.

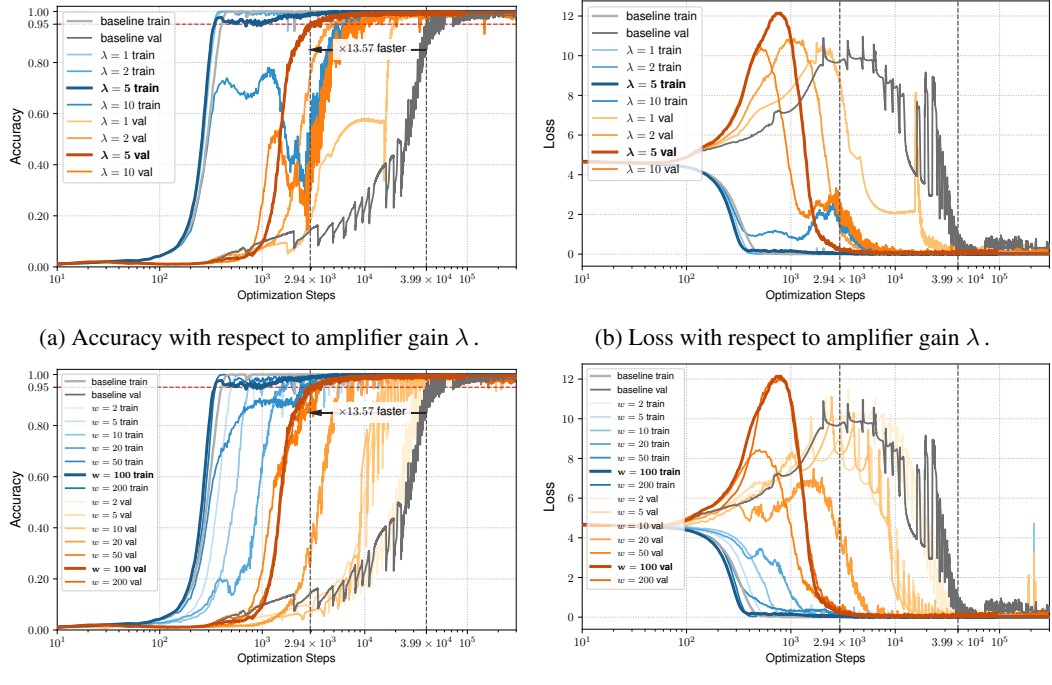

(a) Accuracy with respect to amplifier gain $\lambda$.

(b) Loss with respect to amplifier gain $\lambda$.

(c) Accuracy with respect to window size $w$.

(d) Loss with respect to window size $w$.

Figure 8: **Acceleration of delayed generation with GROKFAST-MA.** The amount of acceleration relies on the two hyperparameters, amplifier gain $\lambda$ and window size $w$. Each hyperparameter has a sweet spot; increasing one arbitrarily does not guarantee faster acceleration. Figures 8a and 8b use $w = 100$ except the baseline. Figures 8c and 8d use $\lambda = 5$ except the baseline.

## 3.4 IMDB

Finally, we train a 2-layer LSTM (Hochreiter & Schmidhuber, 1997) network for sentiment analysis in the IMDb dataset (Maas et al., 2011) under the grokking phenomenon (Liu et al., 2022b). Figure 7 compares the baseline with the model trained with an optimizer modified by Algorithm 1 with $\alpha = 0.98$, $\lambda = 2.0$, and $\text{wd} = 10.0$. We visualize the convergence speed and quantitatively compare the best validation loss/accuracy. This experiment section suggests that GROKFAST generally boosts performance and convergence speed in diverse tasks under the grokking phenomenon.

## 4 GROKFAST WITH MOVING AVERAGE GRADIENT FILTER

### 4.1 REINTERPRETATION AND GENERALIZATION OF MOMENTUM IN OPTIMIZERS

Close inspection reveals subtle resemblance between our GROKFAST-EMA filter and a momentum hyperparameter used in gradient-based optimizers. Specifically, the lines 7-8 of Algorithm 1 take a form similar to Nesterov's momentum, except for the small difference in the time of application: in contrast to Nesterov's momentum, where the momentum is applied *after* the optimizer like in NAdam (Dozat, 2016), our filter is applied *before* the parameter update calculation.

Momentum hyperparameters for gradient-based optimizers are first introduced to smooth out noisy parameter updates generated from mini-batch training (Rumelhart et al., 1986). In contrast, our development of GROKFAST stems from frequency-domain interpretation of the delayed generalization in Section 1. The similarity between GROKFAST-EMA and momentum hypaerparameters, therefore, implies an alternative interpretation of momenta in gradient-based optimizers—they stabilize the training under stochastic gradient-based optimizers *by amplifying the low-frequency component of the parameter updates*. Since a momentum can be seen as a low-pass filter with window of size-1, our new interpretation allows us to generalize the momentum hyperparameter to various types of low-pass

filters of nontrivial $(> 1)$ window size. In this paper, we show one simple extension: GROKFAST with moving average filter.

## 4.2 GROKFAST WITH MOVING AVERAGE FILTER

We devise a simple extension to GROKFAST using a low-pass filter with a nontrivially-sized $(> 1)$ window, i.e., a windowed moving average (MA) filter $h(t)$ over a fixed-size window of size $w$.

$$h(t) = \frac{\lambda}{w} \Pi \left( \frac{t}{w} - \frac{1}{2} \right) = \begin{cases} \lambda/w, & \text{if } 0 \leq t < w. \\ 0, & \text{otherwise.} \end{cases} \tag{7}$$

The function $\Pi$ stands for the Heaviside Pi function, which has value of one in the interval $[-0.5, 0.5]$ and of zero elsewhere. This filter has only two scalar hyperparameters: the scalar factor $\lambda$ and the window size $w$. As shown in Algorithm 2, we implement this filter $h$ with a fixed-capacity queue $Q$ storing the intermediate parameter updates $u(t)$ into the queue $Q$. The average of the queue $Q$ is the low-pass filtered gradients which is added to the current parameter update at each optimizer step.

## 4.3 EXPERIMENT

We compare the results on the same task with the algorithmic dataset as in Section 3. The training curve of this task is shown in Figure 2 as 'baseline.' Comparing the time to reach the accuracy of $0.95$, the generalization, i.e., the late saturation of the validation accuracy, happens after $\times 97.3$ iterations after the rapid saturation of the training accuracy (the overfitting). Figure 8 shows empirical proof of effectiveness of Algorithm 2, our GROKFAST-MA algorithm on this task. Choosing the hyperparameters from a simple grid search over $\lambda \in \{1, 2, 5, 10\}$ and $w \in \{2, 5, 10, 20, 50, 100, 200\}$, we found that the filter works best when $\lambda = 5$ and $w = 100$. As shown in Figure 2, by elongating the window size of a low-pass filter, our Grokfast-MA algorithm performs even better than the Grokfast-EMA algorithm in with best possible hyperparameters. This suggests new class of optmizers augmented with low-pass filters of nontrivially-sized windows that perform better than conventional optimizers with momentum hyperparameters.

## 5 DISCUSSION

Previous sections demonstrate high effectiveness of our approach. However, few questions are still left unanswered regarding the modified training dynamics and the combined effect with the *weight decay*, which is previously shown to be another important algorithmic factor that governs the grokking effect (Liu et al., 2022b). We devise more experiments to answer these questions:

**Q1. Are both slow and fast gradients necessary?** Our approach is based on our belief that the low-pass filtered gradient updates, or the *slow* gradients, contribute to the generalization. The most obvious question is then: can we *not* use the *fast* gradients and replace the original sequence of gradients with the low-pass filtered components? Using only the slow gradients calculated from a moving average filter in Algorithm 2 is equivalent to using larger, overlap-

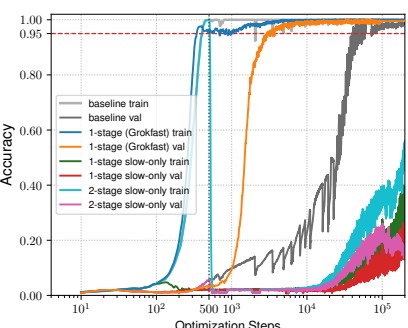

Figure 9: Although *adding* the slow component of the gradients is effective in accelerating grokking, the slow component *cannot* be used alone as a replacement.

ping minibatches. We conduct an experiment with a modified algorithm that replaces the line 8 of Algorithm 2 with $\hat{g}_t \leftarrow \lambda \cdot \text{Avg}(Q)$. Figure 9 shows the result. *1-stage* means that the gradient replacement happens from the beginning of the training, which is set by default, and *2-stage* means that the effect of GROKFAST happens after the model overfits to the training data at iteration $500$. The results clearly reveal that removing the original gradients leads to much slower and unstable training. In conjunction with the result in Figure 8, we can conclude that both the fast and the slow components of the gradients are necessary for faster grokking.

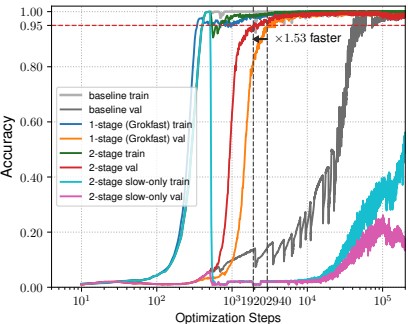

Figure 10: We can further accelerate the grokking effect with a two-staged algorithm, by applying GROKFAST-MA *after* the model is overfitted (after 500 its).

Table 1: Summary of results of Figure 10.

| Name | $\hat{g}_t$ at (A → B) | $\hat{g}_t$ at (B → C) | Iterations at acc ≥ 0.95 |
|---|---|---|---|
| Baseline | $g_t$ | $g_t$ | 39,890 [its] (×1) |
| 1-Stage | $g_t + \lambda \cdot \text{Avg}(Q)$ | $g_t + \lambda \cdot \text{Avg}(Q)$ | 2,940 [its] (×13.57) |
| **2-Stage** | $g_t$ | $g_t + \lambda \cdot \text{Avg}(Q)$ | **1,920** [its] (×20.78) |
| 2-Stage Slow-only | $g_t$ | $\lambda \cdot \text{Avg}(Q)$ | Not converged |

**Q3. Synergistic effect with weight decay.** Besides from our gradient filtering approach, the authors of Omnigrok (Liu et al., 2022b) have suggested that the weight decay hyperparameter is a critical determinant of the grokking phenomenon. According to the report, the grokking phenomenon appears and even becomes faster when the weight decay becomes larger. We, therefore, conduct additional experiments to find out how these two approaches affect the model when applied together. The results are summarized in Figure 11. Compared with the result from GROKFAST-MA with no weight decay (orange), applying the weight decay (red) generally yields even faster generalization. The maximum acceleration appears at $\text{wd} = 0.01$ with ×3.72 faster generalization than GROKFAST-MA with no weight decay. We choose this result of ×50.49 faster grokking to be our main demonstration in Figure 2a. Interestingly, Figure 11 also reveals that applying the same weight decay with no GROKFAST-MA (brown) makes the training unstable. The results demonstrates that applying our gradient filtering and setting up a proper weight decay together gives synergistic benefits.

**Q2. Exploiting state transition in the training of a model under grokking.** We can alternatively interpret the training dynamics of a model under the grokking phenomenon as a *state transition*. In this viewpoint, the model sequentially goes through three distinct stages: (A) *initialized*, where both training and validation losses are not saturated, (B) *overfitted*, where the training loss is fully saturated but the validation loss is not, and (C) *generalized*, where both losses are fully saturated. In the experimental setting of Figure 8, state transition of A → B happens roughly after iteration 500. This interpretation allows us to try out a staged strategy for optimization, where different algorithms are applied to the model during the two transition phases A → B (from iteration 0 to 499) and B → C (after iteration 500) as described in Table 1. Figure 10 and Table 1 summarize the result of the experiment. As the results show, we can accelerate the grokking effect further by ×1.53 by separating the training stage of the model and applying GROKFAST-MA only after the model becomes overfitted, suggesting an adaptive optimizer.

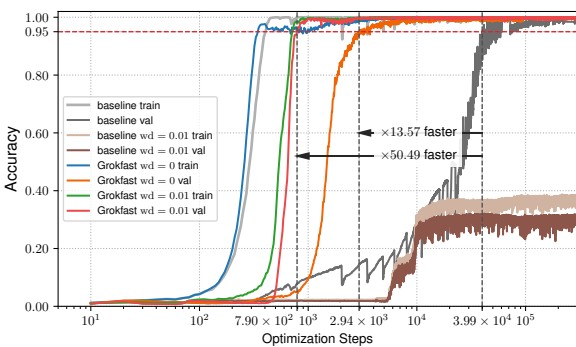

Figure 11: The acceleration effect of GROKFAST-MA is greatly enhanced when accompanied with appropriate value of weight decay. However, the weight decay alone not always yield beneficial results.

### 5.1 VISUALIZING TRAJECTORIES

In this final section, we elaborate on our *state transition interpretation* of grokking introduced in Section 5. Our signal space model of the training dynamics allows us to interpret the training of a model as a random drift of the state in the parameter space. To visualize the dynamics, we collect all the 423k parameters of the Transformer decoder (Vaswani et al., 2017) used in the experiment in Figure 2 for all iterations, and conduct the PCA to obtain the most salient projection of the parameter space. The sequence of evolving models are projected onto the space as in Figure 13a.

Regarding state transition interpretation of grokking, we can observe the followings: First, Figure 12 suggests that, in the baseline setup, the model drifts through a significantly longer pathway from the overfitting (state B, 500 steps) to its full generalization (state C, 300k steps), compared to the initial state (state A, 0 steps) to the overfitting state (state B). However, under GROKFAST, the ratio between the two distances $\overline{\text{AB}}$ and $\overline{\text{BC}}$ in the parameter space becomes more even. This is further acknowledged by Figure 12b and Table 2 showing the distances between the models at each state.

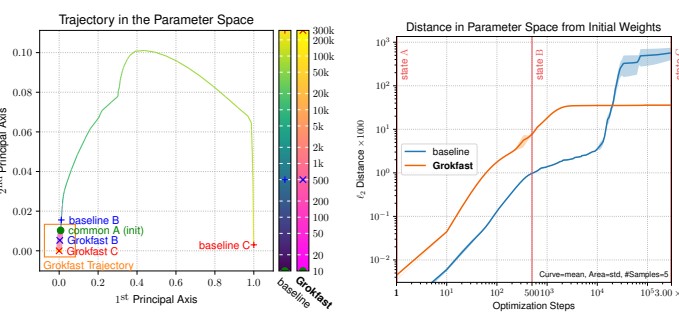

(a) Parameter trajectories.  (b) Deviation from initial weights.

Figure 12: Trajectories of model parameters from experiments of Figure 2 projected onto two principal axes of the PCA of the intermediate model parameters of the baseline. The two models travel along distinct pathways in the parameter space with different pace.

Table 2: Parameter space distances between the intermediate models of experiments in Figure 2. Each state corresponds to each of the markers of Figure 12. Average and standard deviations of five instances are shown. GROKFAST converges to a nearer point in the parameter space, and the baseline model travels longer to reach the final state.

| State Pair | $\ell_2$ Distances ($\times 1000$) | |
| --- | --- | --- |
| | Baseline | GROKFAST |
| $\overline{\text{AB}}$ | $0.97 \pm 2.2$ | $7.7 \pm 0.42$ |
| $\overline{\text{BC}}$ | $563.7 \pm 199.4$ | $15.3 \pm 0.84$ |
| $\overline{\text{AC}}$ | $570.7 \pm 199.8$ | $35.8 \pm 0.98$ |

Moreover, Table 2 suggests that the distances $\overline{\text{AC}}$ between the initial and the final state becomes much ($\times 16$) shorter with our GROKFAST algorithm. Although the generalization accuracy, the training accuracy, the training loss, and the validation loss at the final state (state C) are similar in both the baseline and GROKFAST as showcased in Figure 2, we cannot simply say that the states C of baseline and of GROKFAST belong to the same network state. Likewise the state B of the baseline and of GROKFAST are different. Figure 12b shows average deviation of parameter weights from the initialization point during training of the model under grokking phenomenon. Interestingly, at achieving overfitting at state B, the model under our algorithm deviates $\times 8$ further from the initial point than the baseline does, with $\times 5$ smaller standard deviation in distances from the initial state A. This suggests that although both algorithms exhibit overfitted behavior at state B, intermediate model instances at these states form distinct set of parameters with possibly different topologies. These observations support our interpretation to regard the grokking phenomenon as a state transition between at least three distinct states. The role of GROKFAST is then to provide supervision towards an alternative optimum much nearer from the initial points than the baseline optimizer.

Lastly, the model trained with our GROKFAST algorithm shows hundredfold smaller variances of the distances than the baseline as claimed in Table 2. This implies that training under GROKFAST algorithm is much more deterministic than under typical first-order optimizers. This is possibly related to the similarity between the low-pass filtered gradients from small minibatches with normal gradients from larger minibatches. However, we have also demonstrated in Section 5 that using only the slow, more deterministic component of gradients and completely neglecting the original gradients lead to instability. Therefore, further investigation is needed to find out the source and the role of this determinism from our GROKFAST algorithm, and the reason of its benefits when jointly applied with the faster, more stochastic gradients from baseline optimizers.

## 5.2 LIMITATIONS

Although Algorithm 2 shows highly effective results, it requires $w$ times more memory to store all the previous gradients, limiting its utilization. Replication of the model parameters also makes the training slower; using $w = 100$, the training time per iteration is increased by $\times 2.4$ measured with a single 1080 Ti GPU. Still, the reduction of wall clock time before the delayed generalization of the results in Figure 11 is $\times 20.5$, which is also a notable reduction of time. Though the computation time does not scale linearly with the memory requirements, Algorithm 2 is not generally applicable to the larger models. However, performing better than EMA filters in terms of number of training iterations, our results suggest a sweet spot of optimal window size between 1 (EMA) and 100 (MA). We leave this search for the optimal filters for optimizers for future work.

## 6 RELATED WORK

**Grokking.** The recently discovered grokking phenomenon (Power et al., 2022) signifies one possibility of overparameterized neural networks generalizing (and reasoning) beyond memorization

of the given dataset. Most of the works, thereafter, focus on identifying its mechanism. Recent studies associated grokking with a double descent phenomenon in the DNN mapping geometry training dynamics (Humayun et al., 2023), the speed of pattern learning (Davies et al., 2023), and the sizes of models and datasets (Huang et al., 2024), wherein the validation error initially increases and then decreases with the expansion of model parameters (Nakkiran et al., 2021; Belkin et al., 2019). To investigate the internal roles of each model component during grokking, Nanda et al. (2023) employed mechanistic interpretability, a technique from the XAI domain, and revealed that grokking may not occur abruptly but rather exhibits an internal progress measure. Their assertion posits that the model captures slow, generalizing patterns, underscoring the critical role of proper optimization. Interestingly, while weight decay amplifies the double descent effect (Pezeshki et al., 2022), it contributes to enhanced generalization in grokking scenarios (Power et al., 2022). Liu et al. (2022b) found more examples of grokking in various tasks and analyzed their training mechanism through loss landscape analysis. Thilak et al. (2022) found a similarity between grokking and the slingshot mechanism in adaptive optimizers. Barak et al. (2022) argued that optimizers reach delayed generalization by amplifying sparse solutions through hidden progress. Regularizers such as weight decay (Nanda et al., 2023) and the choice of the optimizer (Liu et al., 2022a) are highlighted as important factors in training a model that groks. Our work is a continuation of this discussion by providing a generalizable tool for the practical study of the grokking phenomenon. Through our discussion, we suggest a state transition model of the grokking and visualize the trajectory of the model weights in the parameter space during training.

**Optimization techniques.** At the core of the study of grokking lies optimization techniques (Thilak et al., 2022). Studies have shown that generalization patterns of the model vary significantly depending on various optimization methods (Power et al., 2022; Gromov, 2023). Power et al. (2022) demonstrated that various factors related to optimization, such as (mini-)batch training (Li et al., 2014), the choice of optimizer, weight decay (Loshchilov & Hutter, 2018), noise injection (Zur et al., 2009), dropout (Srivastava et al., 2014), and learning rate, influence the model's grokking pattern. Nanda et al. (2023) argued that grokking does not occur without proper regularization. Further, they demonstrated that techniques such as weight decay, L2 norm, and dropout induce grokking, but L1 norm does not. On the other hand, Thilak et al. (2022) argued that grokking *can* occur without explicit regularization, attributing this to the optimizer's "visible slingshot mechanism" acting as an implicit regularizer. Liu et al. (2022a) suggested using a larger learning rate for the input embedding segment, facilitating unified learning of the generalization pattern. Unlike these revisiting of the known training techniques, we started from a state space model and a dual domain of the training dynamics. This led us to develop an optimizer augmentation algorithm, GROKFAST, that can be applied to any existing first-order optimizers to accelerate the grokking effect for practical usage.

## 7 CONCLUSION

Our reinterpretation of the deviation of each model parameter into a random signal over training iteration allows us to separate gradient updates into fast-varying and slow-varying components. By amplifying the latter with low-pass filtering, we can bring forward the moment of sudden late generalization, i.e., grokking, reducing the number of required training iterations by up to $\times 50$. Our comprehensive experiments and analyses suggest that our state space interpretation and the frequency representation of the training dynamics is useful for studying the grokking phenomenon. Further discussions have revealed that momentum hyperparameters in optimizers can be regarded as low-pass gradient filters with size-1 windows. Our GROKFAST extends the momentum to general low-pass filters with nontrivially sized windows, bridging the gap between machine learning optimization and classical signal processing literature, suggesting a new class of optimizers based on the well-studied field of signal processing.

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

## A  FREQUENCY RESPONSES OF THE PARAMETER UPDATES UNDER GROKFAST

In Section 1, we have assumed that under a first-order optimizer $u(g(t), t)$, amplifying the low-frequency components of the gradient signal $g(t)$ of an arbitrary parameter $\theta(t)$ over a discrete timestep $t$ has the same effect of amplifying the low-frequency component of the parameter updates $u(t) = u(g(t), t)$. This section mathematically elaborates on the effect of gradient filters $h(t)$ to the parameter update signals $u(t)$ in the most frequently-used type of optimizers: SGD with momentum. For more complicated optimizers such as Adam (Kingma & Ba, 2014) and AdamW (Loshchilov & Hutter, 2018) which cannot be interpreted as linear systems, we take an indirect approach: we show that filtering $g(t)$ instead of $u(t)$ works as it can be seen in the experiments in Section 3.

Stochastic gradient descent with optional momentum term is the simplest and the most widely used optimization algorithm in the deep learning communities. Here, the parameter update $u(t) = \theta(t+1) - \theta(t)$ of a parameter $\theta(t)$ at timestep $t$ and its intermediate momentum $m(t)$ is defined as:

$$m(t) = \mu m(t-1) + (1-\tau)g(t), \tag{8}$$
$$u(t) = -\eta m(t), \tag{9}$$

where $\mu$ is the scalar momentum, $\tau$ is the dampening constant for the momentum, and $\eta$ is the learning rate. This class of optimizers can be thought of as linear systems with state $m(t)$ that receives an input $g(t)$ to produce an output $u(t)$.

To compare the difference between the frequency responses of the parameter update $u(t)$ and of the modified update $\hat{u}(t)$ in equation equation 4, we can think of an equivalent filter $\hat{h}(t)$ defined to satisfy the following relationship in addition to equations equation 3 and equation 4:

$$\hat{u}(t) = u(\hat{g}(t), t) = u(g(t) + h(t) * g(t), t) = u(t) + \hat{h}(t) * u(t). \tag{10}$$

From our assumption of the linear time-invariant, scalar filters $h(t)$ and the linear optimizer, we can deduce the equivalence between $h(t)$ and $\hat{h}(t)$. The following theorem is a generalized claim that applies to any SGD-based first-order optimizers including Nesterov's momentum.

**Theorem A.1.** *Let $g(t)$ be a scalar signal defined over a discrete time $t \in \{0, 1, \ldots, T\}$. Let $h(t)$ be a univariate time-invariant filter defined over the same domain $t$. A linear optimizer $O$ is defined as:*

$$x(t) = Ax(t-1) + Bg(t), \quad t > 0, \tag{11}$$
$$u(t) = Cx(t) + Dg(t), \qquad t \geq 0, \tag{12}$$

*with scalar coefficients $A, B, C$, and $D$, and $x(0) = g(0)$. The output of the system $u(t)$ is, therefore, a function of $g(t)$ and $t$, i.e., $u(t) = u(g(t), t)$. Let the modified input $\hat{g}(t)$, the modified output $\hat{u}(t)$, and the equivalent filter $\hat{h}(t)$ be defined to satisfy the equations equation 3 and equation 10. Then,*

$$\hat{h}(t) = h(t), \tag{13}$$

*for $t \in \{0, 1, \ldots, T\}$.*

*Proof of Theorem A.1.* For simplicity, we first adopt discrete-time Fourier transform over $t \in \mathbb{Z}$. That is, we assume that the signals are defined across every positive and negative integer $t$. Since the value of $u(t)$ can be defined arbitrarily outside the interval $[0, T]$ without modifying the optimization algorithm, we can manually assign $g(t)$ and $x(t)$ for $t \notin [0, T]$ as:

$$g(t) = 0 \qquad t \notin [0, T], \tag{14}$$
$$x(t) = \begin{cases} A^t(1 - B)g(0) & t < 0, \\ A^{t-T}x(T) & t > T. \end{cases} \tag{15}$$

Then, equations equation 11 and equation 12 hold for $t \notin [0, T]$. Note that to make the optimizer $O$ stable, the scalar coefficient $A$ should satisfy $0 < A < 1$. Therefore, the signals $g(t)$ and $x(t)$ are well-defined.

Consider a discrete-time Fourier transform $\mathcal{F}$ defined as:

$$\mathcal{F}\{f(t)\}(\omega) = \sum_{t=-\infty}^{\infty} f(t)e^{-i\omega t}. \tag{16}$$

In the frequency domain, with $G(\omega) = \mathcal{F}\{g(t)\}$, $U(\omega) = \mathcal{F}\{u(t)\}$, and $X(\omega) = \mathcal{F}\{x(t)\}$, the optimizer $O$ can be equivalently represented as:

$$X(\omega) = Ae^{-i\omega}X(\omega) + BG(\omega), \tag{17}$$
$$U(\omega) = CX(\omega) + DG(\omega). \tag{18}$$

We can obtain the transfer functions $H_{\text{in-state}}$ and $H_{\text{in-out}}$ that converts $G$ to $X$ and then to $U$:

$$H_{\text{in-state}}(\omega) := \frac{X(\omega)}{G(\omega)} = \frac{B}{1 - Ae^{-i\omega}}, \tag{19}$$

$$H_{\text{in-out}}(\omega) := \frac{U(\omega)}{G(\omega)} = C\frac{X(\omega)}{G(\omega)} + D = \frac{BC}{1 - Ae^{-i\omega}} + D. \tag{20}$$

If the input $g(t)$ is filtered with a convolutional filter $h(t)$ and then added to itself as equations equation 3 and equation 5, the state $x(t)$ and the output $u(t)$ of the optimizer $O$ is changed accordingly while keeping equations equation 11 and equation 12 hold. We denote $\hat{x}(t)$ and $\hat{u}(t)$ as the modified state and output of the system and $\hat{X}(\omega)$ and $\hat{U}(\omega)$ as their spectra. If the filter $h(t)$ is causal, that is $h(t) = 0$ for $t < 0$, then we can similarly let $\hat{x}(0) = \hat{g}(0)$ and replace $x(t)$ and $u(t)$ with $\hat{x}(t)$ and $\hat{u}(t)$ in equations equation 14 and equation 15 to define an IIR system suitable for the infinite-window discrete-time Fourier transform $\mathcal{F}$:

$$\hat{X}(\omega) = Ae^{-i\omega}\hat{X}(\omega) + B\hat{G}(\omega), \tag{21}$$
$$\hat{U}(\omega) = C\hat{X}(\omega) + D\hat{G}(\omega). \tag{22}$$

Since the coefficients of the linear systems are the same, the transfer functions are identical:

$$\hat{H}_{\text{in-state}}(\omega) := \frac{\hat{X}(\omega)}{\hat{G}(\omega)} = \frac{B}{1 - Ae^{-i\omega}} \qquad \equiv H_{\text{in-state}}(\omega), \tag{23}$$

$$\hat{H}_{\text{in-out}}(\omega) := \frac{\hat{U}(\omega)}{\hat{G}(\omega)} = \frac{BC}{1 - Ae^{-i\omega}} + D \equiv H_{\text{in-out}}(\omega). \tag{24}$$

From equation equation 5, the transfer function of the filter $H_{\text{amp}}(\omega)$ is:

$$H_{\text{amp}}(\omega) = \frac{\hat{G}(\omega)}{G(\omega)} = 1 + H(\omega), \tag{25}$$

where $H(\omega) = \mathcal{F}\{h(t)\}$. The equivalent post-filter $\hat{h}(t)$ defined by equation equation 10 gives another transfer function between the outputs $u(t)$ and $\hat{u}(t)$ of the system:

$$\hat{H}_{\text{amp}}(\omega) = \frac{\hat{U}(\omega)}{U(\omega)} = 1 + \hat{H}(\omega). \tag{26}$$

From equations equation 20 and equation 24, we have:

$$\hat{H}_{\text{amp}}(\omega) = \frac{\hat{U}(\omega)}{U(\omega)} = \frac{\hat{G}(\omega)}{G(\omega)} = H_{\text{amp}}(\omega). \tag{27}$$

Therefore, we get:

$$\hat{H}(\omega) \equiv H(\omega). \tag{28}$$

This completes the proof. □

In other words, applying any filter $h(t)$ to the sequence of gradients $g(t)$ is equivalent to the same filter $h(t)$ applied to the parameter update $u(t)$ for any linear optimizer $O$. This implies that a low-pass *gradient* filter $h(t)$ guarantees the same low-pass property in the modified parameter update signal $\hat{u}(t)$. In many off-the-shelf autograd packages such as PyTorch (Paszke et al., 2019), filtering the gradients is easier and more straightforward than filtering the intermediate parameter updates. The former only adds a few more lines to the outermost application code[1], whereas the latter requires full implementation of the dedicated optimizer object. Note that the above proof holds regardless of the design of the filter $h(t)$ unless there exists a one-to-one correspondence between $H_{\text{amp}}$ and $H$.

The followings are direct consequences of the Theorem A.1.

**Proposition A.2** (SGD with momentum). *Let $t \in \{0, 1, \ldots, T\}$ be a discrete timestep. Let $g(t)$ be a sequence of gradients of a parameter $\theta$ sampled from a stochastic machine learning framework $g(t) \sim M(\theta(t), t)$ and a stochastic gradient descent optimizer $O(\mu, \tau, \eta)$ with a parameter update function $u(g(t), t) = u(t) = \theta(t + 1) - \theta(t)$, a momentum $\mu$, a damping constant $\tau$, and a learning rate $\eta$. The parameter update $u(t)$ is, therefore, defined as:*

$$m(t) = \mu m(t - 1) + (1 - \tau)g(t), \tag{29}$$

$$u(t) = -\eta m(t), \tag{30}$$

*with a scalar momentum term $m(t)$ for each parameter $\theta$ with $m(0) = g(0)$. Let $h(t)$ be a scalar, time-invariant, convolutional gradient filter. Let the modified input $\hat{g}(t)$, the modified output $\hat{u}(t)$, and the equivalent filter $\hat{h}(t)$ be defined to satisfy the equations equation 3 and equation 10. Then,*

$$\hat{h}(t) = h(t), \tag{31}$$

*for $t \in \{0, 1, \ldots, T\}$.*

*Proof of Proposition A.2.* Let $A = \mu$, $B = 1 - \tau$, $C = -\eta$, and $D = 0$. By Theorem A.1, equation equation 31 holds. □

---

[1]See our implementation in the Supplementary Material.

**Proposition A.3** (SGD with Nesterov's momentum). *Let $t \in \{0, 1, \ldots, T\}$ be a discrete timestep. Let $g(t)$ be a sequence of gradients of a parameter $\theta$ sampled from a stochastic machine learning framework $g(t) \sim M(\theta(t), t)$ and a stochastic gradient descent optimizer $O(\mu, \tau, \eta)$ with a parameter update function $u(g(t), t) = u(t) = \theta(t + 1) - \theta(t)$, a momentum $\mu$, a damping constant $\tau$, and a learning rate $\eta$. The parameter update $u(t)$ is, therefore, defined as:*

$$m(t) = \mu m(t-1) + (1-\tau)g(t), \tag{32}$$
$$u(t) = -\eta(g(t) + \mu m(t)), \tag{33}$$

*with a scalar momentum term $m(t)$ for each parameter $\theta$ with $m(0) = g(0)$. Let $h(t)$ be a scalar, time-invariant, convolutional gradient filter. Let the modified input $\hat{g}(t)$, the modified output $\hat{u}(t)$, and the equivalent filter $\hat{h}(t)$ be defined to satisfy the equations equation 3 and equation 10. Then,*

$$\hat{h}(t) = h(t), \tag{34}$$

*for $t \in \{0, 1, \ldots, T\}$.*

*Proof of Proposition A.3.* Let $A = \mu$, $B = 1 - \tau$, $C = -\eta\mu$, and $D = -\eta$. By Theorem A.1, equation equation 34 holds. □

## B  TASK DETAILS

For completeness, this section summarizes the implementation details of each task dealt in Section 3. The readers can also consult our official implementation in PyTorch (Paszke et al., 2019).

### B.1  BINARY OPERATION (ALGORITHMIC DATA)

This is the description of algorithmic data used throughout the manuscript. Following the first report on the grokking phenomenon (Power et al., 2022), we demonstrate our acceleration algorithms with a binary operation $x \cdot y \pmod{p}$, with $p = 97$. The network is a two-layer decoder-only Transformer (Vaswani et al., 2017) with hidden dimension of 128 and 4 heads in its attention. The positional embedding has length of 5, and GELU (Hendrycks & Gimpel, 2016) and layer normalization (Lei Ba et al., 2016) is used throughout the network. After the Transformer blocks, the output is fed into a layer normalization and a linear output layer to return logits. We use cross entropy loss to train the network and an Adam (Kingma & Ba, 2014) with betas $(\beta_1, \beta_2) = (0.9, 0.98)$, a constant learning rate of $10^{-3}$, batch size of 512, and linear learning rate warmup schedule over the first 10 iterations.

### B.2  MNIST

We train a three-layer MLP with hidden width of 200 and ReLU activations for the MNIST classification task (Deng, 2012). Under $\times 8$ larger weight initialization than Kaiming initialization (He et al., 2015), the network is known to exhibit the grokking phenomenon (Liu et al., 2022b). The network receives flattened grayscale images of size $28 \times 28$ and outputs 10-dimensional logits to calculate MSE losses between one-hot encoded labels. We use the batch size of 200 and trained with an AdamW optimizer (Loshchilov & Hutter, 2018) with a constant learning rate of $10^{-3}$ until $10^5$ training iterations. We use a smaller subset of 1000 images from training images to train the network in order to simulate overfitting environment. All the other hyperparameters are set by default.

### B.3  QM9

To demonstrate the effectiveness of our algorithm on a graph convolutional neural network, we use QM9 small molecules dataset (Ruddigkeit et al., 2012; Ramakrishnan et al., 2014) to estimate the isotropic polarizability. Our Graph ConvNet has two graph convolution layers with input channel of 11 (QM9 edge features), output channel of 16, and hidden channel of 32. Each graph convolution is followed by a ReLU. Each convolution layer consists of two linear layers with an internal ReLU activation with hidden channel of 32. The output of the Graph ConvNet is a global average pooling, followed by a two-layer MLP with a ReLU and hidden channel of 32. To simulate the overfitting environment, we use the first 100 samples from the data. The data is again randomly split into train

and validation sets with 50:50 size ratio. We use batch size of 32, an AdamW optimizer (Loshchilov & Hutter, 2018) with a constant learning rate of $10^{-3}$. The network is initialized with weights $\times 3$ larger than that of Kaiming initialization (He et al., 2015) and trained for 50k iterations.

### B.4 IMDB

For IMDb dataset (Maas et al., 2011), we use LSTM (Hochreiter & Schmidhuber, 1997) with two layers, embedding dimension of 64, hidden dimension of 256, and vocabulary size of 1001, including the padding token. The network is followed by a single fully connected layer with output dimension of 1 with sigmoid activation to classify the positive/negative sentiment of each review string. The dataset was preprocessed by tokenizing the 1000 most frequent words from the review. The list of integer tokens are padded by zeros to form an array of reviews with the same length of 500. The network was trained by a binary cross entropy loss and an AdamW optimizer (Loshchilov & Hutter, 2018) with learning rate of $3 \times 10^{-4}$ and batch size of 50. We trained the model with the first 1000 rows from the dataset, split into train and validation sets with 75:25 size ratio. We stopped the training at 10k iterations as shown in Figure 7.

## C IMPLEMENTATION GUIDE

We have argued that our implementation of Algorithm 2 and 1 costs only a few additional lines of code. We demonstrate this by presenting the exact code we developed with the PyTorch (Paszke et al., 2019) autograd package. The readers who are interested can also consult our official implementation in the Supplementary Material.

Algorithms 2 and 1 are implemented as follows:

```python
# Grokfast-MA (Algorithm 1)
def gradfilter_ma(
    m: nn.Module,
    grads: Optional[Dict[str, deque]] = None,
    window_size: int = 100,
    lamb: float = 5.0,
    filter_type: Literal['mean', 'sum'] = 'mean',
    warmup: bool = True,
) -> Dict[str, deque]:
    if grads is None:
        grads = {n: deque(maxlen=window_size) for n, p in
            m.named_parameters() if p.requires_grad}

    for n, p in m.named_parameters():
        if p.requires_grad:
            grads[n].append(p.grad.data.detach())

            if not warmup or len(grads[n]) == window_size:
                if filter_type == "mean":
                    avg = sum(grads[n]) / len(grads[n])
                elif filter_type == "sum":
                    avg = sum(grads[n])
                else:
                    raise ValueError(f"Unrecognized filter_type
                        {filter_type}")
                p.grad.data = p.grad.data + avg * lamb

    return grads

# Grokfast (Algorithm 2)
def gradfilter_ema(
    m: nn.Module,
    grads: Optional[Dict[str, torch.Tensor]] = None,
    alpha: float = 0.98,
    lamb: float = 2.0,
) -> Dict[str, torch.Tensor]:
```

```
35    if grads is None:
36        grads = {n: p.grad.data.detach() for n, p in m.named_parameters()
              if p.requires_grad}
37
38    for n, p in m.named_parameters():
39        if p.requires_grad:
40            grads[n] = grads[n] * alpha + p.grad.data.detach() * (1 - alpha)
41            p.grad.data = p.grad.data + grads[n] * lamb
42
43    return grads
```

This helper method can be applied to any optimization framework involving the autograd package by inserting a single line between the calculation of the gradients and the optimizer call as follows:

```
1    # ... any initialization code before starting the training loop.
2    grads = None
3
4    # Training loop.
5    for batch in dataloader:
6        model.zero_grad()
7        output = model(batch)
8        loss = criteria(output)
9
10       # Calculate the gradients.
11       loss.backward()
12
13       # Option 1: Grokfast (has argument alpha, lamb)
14       grads = gradfilter_ema(model, grads=grads, alpha=alpha, lamb=lamb)
15       # Option 2: Grokfast-MA (has argument window_size, lamb)
16       # grads = gradfilter_ma(model, grads=grads, window_size=window_size,
              lamb=lamb)
17
18       # Call the optimizer.
19       optimizer.step()
20
21       # ... any additional logging codes.
```

Note that line 2 and line 14 in the code above are the only modification we made.

## D    TIME AND MEMORY REQUIREMENTS

This section delivers additional demonstration of the efficiency of our GROKFAST algorithm. As we have argued in Section 5.2, the additional computational burden from our augmentation is compensated by the larger-scale acceleration of the delayed generalization. The additional cost of VRAM memory is also negligible compared the baseline. The time and the memory requirements in the Tables 3 through 6 are measured with a single GTX 1080 Ti GPU.

## E    MORE VISUALIZATION

We finally provide more visualization in addition to Section 5 in order to understand the training dynamics under our GROKFAST algorithm. Figure 13 shows five more runs from the same experiments in Figure 2 and 12 with different seeds. We saved all the 423k parameters of the Transformer decoder (Vaswani et al., 2017) from every training iteration, likewise in Figure 12. The parameters of a model checkpoint from each run at each iteration are reshaped into a single long vector. The vectorized parameters are then normalized by subtracting them by the model's initialized weights. This way, we can align the trajectories by centering the initial states (state A) of all the experiments at the origin. The sequence of parameter differences of the ten runs from the two algorithms, i.e., baseline and GROKFAST, forms a tensor of shape ((Number of Runs) · (Number of Iterations)) $\times$ (Number of Parameters) = ((Number of Sampled Iterations) $\times$ 422784). From this we perform

Table 3: Quantitative results of GROKFAST with a Transformer decoder trained for the algorithmic data (modular multiplication). The experiments corresponds to that of Figure 2 and 4a & 4b.

| Algorithm | Iterations @ 95% Val. Acc. | Wall Clock Time @ 95% Val. Acc. (s) | VRAM (MB) | Latency Per Iteration (s) |
|---|---|---|---|---|
| Baseline | 39890 | 5984 | 290 | 0.15 |
| GROKFAST-MA | 790 ($\times$ 50.49 $\downarrow$) | 292 ($\times$ 20.49 $\downarrow$) | 458 | 0.37 |
| GROKFAST | 910 ($\times$ 43.84 $\downarrow$) | 137 ($\times$ 43.79 $\downarrow$) | 294 | 0.15 |

Table 4: Quantitative results of GROKFAST with an MLP trained for MNIST (Figure 5).

| Algorithm | Iterations @ 95% Val. Acc. | Wall Clock Time @ 95% Val. Acc. (s) | VRAM (MB) | Latency Per Iteration (ms) |
|---|---|---|---|---|
| Baseline | 44022 | 1928 | 196 | 43.8 |
| GROKFAST | 2001 ($\times$ 22.00 $\downarrow$) | 87.8 ($\times$ 21.96 $\downarrow$) | 198 | 43.9 |

Table 5: Quantitative results of GROKFAST with a G-CNN trained for QM9 (Figure 6).

Table 6: Quantitative results of GROKFAST with an LSTM trained for IMDb (Figure 7).

| Algorithm | Minimum Val. Loss | VRAM (MB) | Latency Per Iteration (ms) | Algorithm | Best Val. Acc. | Minimum Val. Loss | VRAM (MB) | Latency Per Iteration (ms) |
|---|---|---|---|---|---|---|---|---|
| Baseline | 0.00659 | 216 | 40.2 | Baseline | 0.84 | 0.517 | 754 | 20.4 |
| GROKFAST | 0.00348 | 216 | 41.4 | GROKFAST | 0.90 | 0.412 | 762 | 21.2 |

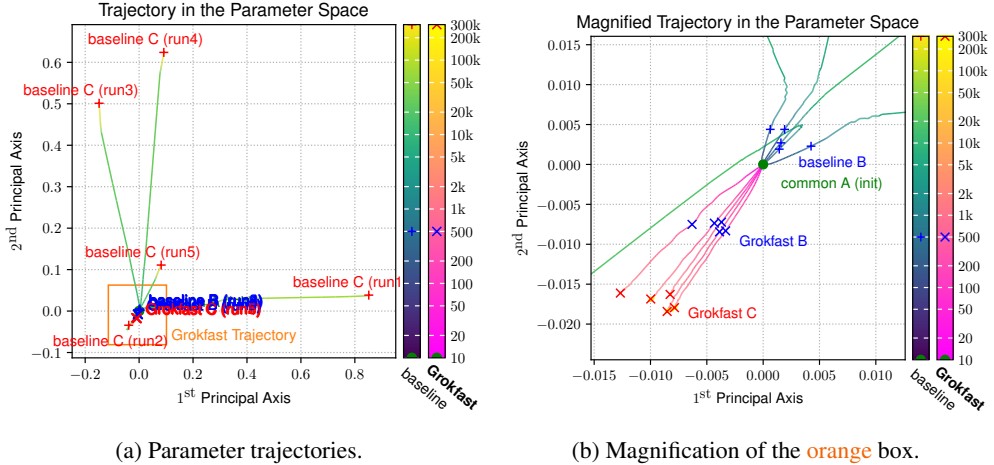

(a) Parameter trajectories.

(b) Magnification of the orange box.

Figure 13: Normalized trajectories of model parameters from five runs of the experiment of Figure 2. The baseline optimization algorithm *without* GROKFAST guide the model to the overfitting states (state B) relatively closer to the initialization states (state A). After reaching the overfitting state, however, the model parameters travel far away to reach the generalization states (state C). GROKFAST instead guide the model parameters to the alternative generalization states (state C), which are much closer to the initialization states (state A).

the PCA to obtain the projection matrix of shape $422784 \times 2$. This matrix projects the parameter differences from each of the model checkpoint onto the two most salient directions of variations. We mark the initialization state (state A), the overfitting state (state B, 500 iterations), and the generalization state (state C) from each run in the two-dimensional plot. The results are Figure 13.

We first notice that the overfitting states (state B) from each of the two optimizers are clearly different. The baseline algorithm without GROKFAST reaches the overfitting states (state B), which are relatively nearer to the initialization states (state A) than those of GROKFAST algorithm. However, as soon as the model overfits, the weights continue to deviate far from the points where overfitting first occured (state B). As a result, the final generalization (state C) happens much far away from the initialized weights (state A). It is notable that the five generalization states (state C) from different instances of the baseline optimizer vary significantly. The difference between the states C of the baseline is much larger than that of the baseline's overfitting states (state B) and that of the states B and C from our GROKFAST algorithm. In contrast, the training dynamics from GROKFAST results in a distinct set of trajectories that lead to the generalization states (state C) much closer to the initial weights.

Moreover, difference within the trajectories from GROKFAST is much smaller than that of the baseline algorithms. This conclusion is also verifiable from Figure 13b and Table 2 in a quantitative manner.

