# OpenReview forum: "Grokfast: Gradient filters for faster grokking"
_ICLR.cc/2025/Conference — ICLR 2025 Conference Withdrawn Submission_

### Official Review · Reviewer_ubaB · 2024-10-21

**Soundness:** 2
**Presentation:** 3
**Contribution:** 2
**Rating:** 5
**Confidence:** 3

**Summary:**

This paper introduces a very interesting artifact in optimizing neural networks, which is Grokking. The authors of this paper believe some low-frequency component of parameter updates is the key to the generalization of model. They thus design an optimizer with a discrete-time low-pass filter (LPF), using EMA and MA techniques to amplify low-frequency components. The method is verified on several basic benmarks and demonstrates faster generalization or higher accuracies on evaluation datasets.

**Strengths:**

* This paper studies a magical phenomenon: grokking. The idea of achieving faster grokking via a well-designed optimizer is highly interesting and attractive.
* The authors connect the optimization with the field of classical signal processing, I acknowledge the potential of this analysis approach.
* The method is simple and easy to implement practically, and it indicates a preliminary advantage of fast generalization in experiments as the authors claimed.
* They try to provide more insights about grokking in the discussion part, which might be helpful to future studies.

**Weaknesses:**

* This paper will make a much better contribution if the authors provide a rigorous mathematical derivation or analysis for presenting intuition about why they believe *amplifying the low-frequency component of $G(\omega)$ accelerates the speed of generalization under grokking phenomenon*.
* It’s not clear enough about how scaling *slow gradients* will benefit or hurt the grokking process. Can the occurrence of the grok phenomenon be slowed down by reducing the low-frequency component of updates? If possible, I recommend the authors add this experiment to verify their hypothesis by using some techniques like high-pass filtering the gradients.
* Although Grokfast performs fairly better than baselines in the experiment, I doubt its effectiveness in training the modern large-scale neural networks since the MA and EMA tricks are now widely used in optimizers, but none of them could beat popular SGDm and AdamW. We don’t know when and whether the grok phenomenon will appear. For example, grokking might never appear in pretraining language model on comprehensive dataset. I am concerned that the conclusion and method might not be able to generalize to all ML tasks.
* Minor issue: The authors mixed up the labels of algorithms in Appendix C.

**Questions:**

* Please answer the issues in the Weakness and point out my potential misunderstandings. I am happy to discuss and enhance my rate.
* Could you add a visualization to analyzing the dynamics of the high-frequency component of parameters and low-frequency component of parameters on some specific datasets?
* It will be good if you also add a simple convergence analysis of Grokfast to provide basic theoretical conclusions.

---

### Official Review · Reviewer_v22w · 2024-10-27

**Soundness:** 2
**Presentation:** 2
**Contribution:** 2
**Rating:** 3
**Confidence:** 4

**Summary:**

The authors investigate the phenomenon of grokking, where the number of update steps required for generalization far exceeds those needed to achieve near-perfect training accuracy. Motivated by this difference in time scales, they propose amplifying the low-frequency components of the series of mini-batch gradients during training to accelerate generalization. They introduce two algorithms: one that adds an exponential moving average of the gradients to the current gradient, and another that adds a simple moving average. Empirical results on several tasks show that, with appropriate hyperparameters, these methods significantly reduce the number of steps needed for generalization compared to the number of steps needed for generalization in a baseline setup.

**Strengths:**

Motivating the use of moving averages in the proposed algorithms as low-pass filters in the frequency domain of the gradient series is an interesting domain-spanning approach.

The authors rigorously validate their algorithms across a variety of datasets and network architectures, demonstrating broad applicability.

**Weaknesses:**

In most setups, grokking can be avoided by selecting appropriate hyperparameters. Liu et al. (2022a) describe grokking as an interesting but rare phenomenon, noting that simply optimizing hyperparameters, such as the learning rate, can significantly speed up training and prevent grokking without requiring special algorithms. While the authors of this paper do acknowledge the importance of optimizer choice and hyperparameters in principle, they emphasize their algorithm’s ability to accelerate generalization with minimal code changes (e.g., lines 19-20, 196-197, 473, 518, 526) and present speedup as their primary goal (lines 14-15). However, given the existing literature, achieving faster training or avoiding grokking through these methods is not particularly surprising. In general, itmight be an interesting question how different hyperparameters and optimizers affect grokking, but the authors do not present their results in this way.


As the authors note, their exponential moving average (EMA) algorithm closely resembles a conventional optimizer that incorporates a momentum hyperparameter. Although the authors also introduce an algorithm utilizing a simple moving average (MA), which demonstrates a slight improvement in generalization speed up (50x for MA compared to 44x  for EMA), the overall impact on training performance appears comparable to that of the exponential moving average (see Figure 2 and Figure 4, panels (a) and (b)). This observation raises the question, why no conventional momentum hyperparameter was investigated and compared to, and leaves the reader with results solely for the unconventional algorithms and no proper comparison.

Overall, the authors provide minimal hyperparameter comparison for the baseline case that does not incorporate any of their proposed algorithms. In particular, they do not investigate the effects of varying the learning rate, which has been shown to significantly influence the grokking phenomenon (Liu et al., 2022a).

Liu et al. (2022a): https://arxiv.org/abs/2205.10343

Minor comment:

line 188: Typo: mentioning window size $w$ as a hyperparameter of the exponential moving average algorithms instead of momentum $\alpha$.

**Questions:**

Why is it presented as the primary goal "to accelerate the generalization of a model under the grokking phenomenon", when the literature suggests that grokking can often be mitigated through proper hyperparameter optimization (Liu et al.)? Especially when a special initialization and a smaller training dataset are needed to see any grokking at all, for example, with MNIST.

Why is there no comparison with an optimizer that employs a conventional momentum hyperparameter? Additionally, why is there no baseline comparison of different learning rates?

---

### Official Review · Reviewer_1pnw · 2024-11-02

**Soundness:** 2
**Presentation:** 3
**Contribution:** 2
**Rating:** 3
**Confidence:** 3

**Summary:**

Grokking is the recently observed phenomenon of delayed generaization, in which the model trained by a gradient-descent-based algorithm first quickly learns the target function on the training set and then, much later, learns to generalize it to the whole domain. The present paper proposes a method of reducing the delay by filtering the gradients. The method is based on the idea of low-pass-filtering. The paper suggests that the delayed generalization is associated with high-frequency oscillations of the gradient, so learning can be streamlined by smoothing the gradient using an exponential or windowed moving average. The paper provides several experiments with synthetic and realistic problems in which the proposed algorithm indeed reduces the generalization delay. After that the authors discuss several properties of their algorithm, in particular the impossibility of entirely removing the fast spectral components of the gradients and a synergy with weight decay. Finally, the paper visualizes the learning trajectories and argues that the proposed method shortens them significantly.

**Strengths:**

The paper demonstrates experimentally that a generalization delay in problems exhibiting grokking can be significantly reduced by simple momentum- or averaging-based modifications of the learning algorithm. I have not seen the effects of momentum and averaging discussed in other papers on grokking (but I'm not an expert in grokking; I have only seen a couple of earlier papers on the topic). This is potentially a useful observation.

The experiments in the paper are performed across several datasets and network architectures.

The paper provides some simple intuiition for why the proposed modifications improve generalization (namely, that they act as low-pass filters).

On the whole, the paper is easy to read.

**Weaknesses:**

1. The approach developed in the paper is motivated by the claim that delayed generalization in grokking is due to high-frequency oscillations of the gradient (line 94). I don't see what confirms this claim. The paper does not seem to cite or present any evidence here.

2. The eficiency of the proposed filtering approach is demonstrated only by experimental evidence. The paper does not make any theoretical analysis beyond the vague generic statement that low-pass filtering should reduce high-frequence oscillations and thus reduce generalization delay.

3. The experiments with synthetic problems (modular multiplication) do confirm a substantial reduction of generalization delay by the proposed method. However, in the case of experiments with real data the situation is not so obvious. For MNIST, the authors seem to use an oddly inefficient three-layer ReLU-MLP that has a very low accuracy of 90%. Their algorithms speeds-up generalization, but the accuracy remains at around 91%. In contrast, the usual accuracy of even basic MLP's on MNIST is 98% or so. I think that it is misleading to omit a discussion of this point in the paper.
In the case of QM9 and IMDB both grokking and the effect of the proposed algorithm are not very pronounced. The proposed filtering seems to be analogous to adding momentum or averaging - commonly used techniques well-known to benefit optimization. The improvement visible in the plots can be attributed to standard improvement from these techniques.

4. The paper contains a theoretical Appendix A, but it only includes theorems that seem to be of relatively technical nature (equivalence between filtering the gradients and the parameter updates) and don't do much to increase confidence in the proposed method. These theorems don't seem to be referenced in the main text.

5. Given its main focus, the paper does not provide sufficient background on phenomenology and previous theoretical studies of grokking. The paper includes Section 6 on related work, but it only relatively superficially mentions previous research, in particular not going into much detail how various factors increase or reduce grokking. I would expect the present paper to discuss much more extensively how its idea of low-pass filtering corresponds to the existing phenomenology (ideally, already in the introduction) and alternative theories of grokking. In particular, I have also seen the paper [1] explaining grokking by the lack of feature learning and claiming an alternative approach to remove grokking by inducing feature learning.

[1] Kumar et al., Grokking as the Transition from Lazy to Rich Training Dynamics, ICLR 2024

6. Grokking appears to be a relatively rare and somewhat artificial phenomenon, uncommon in real-world data and neural networks. Accordingly, whereas a better theoretical understanding of grokking would certainly be appreciated, I'm not sure that algorithms aimed at eliminating grokking are practically very important.

Summarizing, I think that the paper in its present state is not good enough for ICLR in terms of significance, quality, and usefulness for the reader. I think that the paper can be substantially improved by addressing the items indicated above.

**Questions:**

The proposed filtering seems to be analogous to adding momentum or averaging to plain (stochastic) gradient descent. Is there any essential difference between the proposed filtering and these standard modifications? Can we view standard Heavy Ball (GD with momentum) as a low-pass filtered GD? If we replace GD by Heavy Ball, does this eliminate/reduce grokking? Or does your low-pass filtering applied to plain GD create a substantially new momentum-type algorithm that removes generalization delay better than standard Heavy Ball?

---

### Official Review · Reviewer_U18L · 2024-11-05

**Soundness:** 2
**Presentation:** 2
**Contribution:** 2
**Rating:** 5
**Confidence:** 4

**Summary:**

The article studies the grokking phenomenon
in deep learning where the test accuracy improvement
during a training procedure
There is a strong delay relative to the training accuracy.
It shows that reducing the delay is possible
if one can smooth out the fast-varying
components in stochastic gradients.
Numerically, the proposed idea is validated on several
strong benchmarks, showing a significant delay-time reduction.

**Strengths:**

- The main idea is clear and well explained,
and extensive numerical simulations are performed
to study the grokking phenomenon.

**Weaknesses:**

- The moving average in Algorithm 1 to compute hat g_t is
conceptually very close to the Adam method. In the article,
the authors have used this method (AdamW) in the numerical results,
therefore it is unclear whether the effect of moving averaging
is so crucial or not to explain the delay-time reduction.

In this sense, the main message of the article is conceptually not fully convincing.
I suggest the authors to
conduct an ablation study on the same experiments
without using Adam optimizers (e.g. u(g,t) is simply SGD without momentum).

- The writing could be further improved. Certain notation such as
the convolution in eq. 3 is not standard. One would use h*g(t) in this case.

**Questions:**

- Is the theta(t) in eq. 1 in one-dimensional, or is it a vector of multiple dimensions?
I ask since you taking the Fourier transform on u in eq. 2 on their difference.
It seems that theta(t) is in R, and thus you are considering each component
of a gradient descent method separately.
Please explicitly state the dimensionality of theta(t) and explain how the Fourier transform is applied if theta(t) is multi-dimensional.


- When reading ALgorithm 2, it is unclear what it means Avg(Q). It would be better
to explain this in Section 2 rather than in Section 4.
Or you could add a brief explanation of Avg(Q) in the algorithm description in Section 2.

---

### Note · Authors · 2024-11-22

**Comment:**

We deeply appreciate all the reviewers for their thoughtful and constructive feedback. After intense internal discussion, we have concluded that we need a nontrivial amount of change in writing in order to deliver our findings more convincingly, which is not feasible for this time interval we are given. Therefore, we decided to withdraw this submission and seek for improvement based on your considerate feedback. We regret to inform you the decision we have made. Thank you again for the insightful questions regarding this work.

**Withdrawal Confirmation:**

I have read and agree with the venue's withdrawal policy on behalf of myself and my co-authors.